# The Mechanisms of the Transportation Land Transfer Impact on Economic Growth: Evidence from China

Mingzhi Zhang [1], Zhaocheng Li [1], Xinpei Wang [1], Jiajia Li [2,*], Hongyu Liu [1] and Ying Zhang [1]

[1] School of Economics, Institute of Population and Economic Development, Shandong University of Finance and Economics, Jinan 250014, China; zmz915@sdufe.edu.cn (M.Z.); lizhaocheng@mail.sdufe.edu.cn (Z.L.); wangxinpei@mail.sdufe.edu.cn (X.W.); liuhongyu@mail.sdufe.edu.cn (H.L.); 19993702@sdufe.edu.cn (Y.Z.)

[2] Hospitality Management School, Shanghai Business School, Shanghai 201400, China

\* Correspondence: 21180013@sbs.edu.cn

**Abstract:** Accessibility to transportation is a crucial factor for economic growth. Transportation land, defined as the land used to support transportation infrastructure, such as city and inter-city rail, ports, and air travel, is a critical element for constructing transportation facilities and has attracted increasing attention from researchers and policy makers. Transportation land transfer (TLT) is defined as the act by which the state transfers transportation land-use rights to a land user (collective or individual) within a certain period of time as the land owner (all land in China is owned by the state). The land user pays a land-use right transfer fee to the state. This article first reveals the multidimensional effect between TLT and economic growth based on data from China's 30 provinces for 2007–2019. The study found the following. (1) A continuous increase in the availability of transportation land is vital to ensure sustainable economic growth, and the construction of transportation land between adjacent areas has positive spatial spillover effects. (2) These positive effects work through three mechanisms, i.e., increased employment, industrial interactions, and improvements in economic operational efficiency, with a time lag. (3) The positive effects of TLT on economic growth have significant heterogeneous moderating effects on the differences in the economic development stage, the level of industrial structure, and urbanization rate. The study expands the front-end to back-end analysis of land use, provides a reference for countries and regions at different stages of development to promote economic growth using transportation land construction, and presents beneficial insights for governments to efficiently avoid the mismatch of transportation land resources.

**Keywords:** transportation land transfer; economic growth; threshold effect

## 1. Introduction

Transportation is an important supporting resource for national, regional, and urban economic and social development [1]. Therefore, it is important to address the availability of land for transportation services, which is a key element when constructing transportation facilities. Land is a critical factor of production that directly affects national economic development and growth potential [2]. Transportation land is defined as land that supports transportation infrastructure, such as different types of rail, ports, and urban roads; it is an essential type of land resource for optimizing industrial layouts [3], upgrading industrial structures [4], and encouraging urbanization [5]. Transportation land transfer (TLT) is defined as the act by which the state transfers transportation land-use rights to a land user (collective or individual) within a certain period of time as the land owner (all land in China is owned by the state). Transportation infrastructure, supported by transportation land, can effectively encourage the orderly flow of multiple factors of production, improve the productivity of industrial sectors, and determine the layout and development of different industries [4].

Countries in different stages of development have different demands for transportation land. Countries in the early and middle stages of industrialization have accelerated population agglomeration and increased their active economic activities. As such, they

urgently need more transportation land (transportation infrastructure) to meet production and living needs [6,7]. In contrast, for countries in the later stage of the industrialization and primary development stage, the original supply mode of transportation land cannot support further development of urbanization and the smooth transformation and upgrading of the industrial structure. It is vital for these countries to align with the laws of development and reform the mode of transportation land supply to encourage the progressive development of the national economy.

Traffic land has the characteristics of phased demand, which means that the demand for the quantity and type of land for traffic at different development stages is significantly different. Fulfilling that demand may further promote economic growth [6], population aggregation [8], industrialization [9], urbanization [5], changes in land use [10], and other factors [11,12]. Therefore, many governments have expressed interest in phased transportation land supply planning. To meet the challenges of advanced economic development, most countries have implemented actions in the field of transportation. For example, the European Union launched the TRANSPLUS project in 2000 to develop planning tools and best practices to better facilitate the management of future transport demand [13]. Australia established its national land transport network in 2014, and the United States formulated a "Beyond Transport" strategic plan in 2015 [14]. As a developing country, China has engaged in a strategic planning agenda to build a powerful transportation country; this agenda has been included in the "Outline of the 14th Five-Year Plan (2021–2025) for National Economic and Social Development and Vision 2035 of the People's Republic of China." In February 2021, China issued the "Outline of the National Comprehensive Three-dimensional Transport Network".

Transportation land supply plans have been issued in many countries; corresponding research and specific recommendations are needed to support macro-planning and implementation. Past studies in the field of transportation land use have mainly focused on the spatial and temporal patterns and factors influencing the use of transportation land. However, few studies have explored the relationship between provincial TLT and economic growth; this relationship and the internal driving mechanism deserve more extensive studies. In the past 40 years of reform and opening-up, China has experienced different stages of urbanization and changes in its industrial structure, which has led to a significant wealth of accumulated experience to learn from. First of all, as the largest developing country in the world, the economic development policies formulated by China are relevant as valuable references for underdeveloped countries. In addition, since a horizontal analysis of 30 provinces in different stages of economic development and a vertical analysis through threshold effects were conducted in this work, the results can also provide a useful reference for countries in a higher stage of development. Notably, all land in China is owned by the state, which can grant the right to use the land. In the transfer of land-use rights, state-owned land-use rights of certain land plots are transferred from the state to land users for a certain period of time. In this process, the state remains the land owner and is paid land-use fees by the users. Land-use rights can be granted through agreements or via bidding and auctions. Land users can be collective or individual. Although land systems and methods of TLT vary in different countries, the essence is to conduct planning and market transactions to realize highly efficient land use and economic growth. Therefore, the research of this paper is significant on both a general and stage-specific level. In addition, the role of transportation infrastructure emphasizes the links between regions. The research of transportation land planning on provincial level is beneficial to the coordination among cities in the province and promotes the overall development within regions.

In the context of the large increase in the supply of transportation land (transportation infrastructure construction), China's economic growth rate remains strong. However, the questions remain: What is the relationship between the two variables, TLT and economic growth? What are the best ways to identify the relationship between the two variables in the various stages of economic development, industrial structure, and urbanization? Answering these questions is critical for further research and planning. Therefore, it is

important for government agencies across countries to establish a phased decision-making system for transportation land supply.

The goals of this research were: (1) to construct a decision-making framework for transportation land supply considering the stage of economic development, industrial structure, and urbanization; and (2) to provide a reference for decision making with respect to the supply of transportation land. To achieve the objectives above, we analyze and test the spatial spillover effect, the mechanism of action, and the threshold effect of TLT on economic growth by applying a spatial econometric model, an intermediary effect model, and a threshold effect model, based on provincial transportation land transfer data and other economic data.

The contribution of this study lies in three main areas. (1) It is the first known study of the dynamic effect of TLT on economic growth from the perspectives of employment, industrial interaction, and economic operation efficiency. This work enriches our theoretical and empirical understanding of the economic effect created by TLT. (2) To enhance the reliability of the conclusions, two spatial weight coefficients are used to comprehensively evaluate the spatial effect of TLT. (3) We apply the perspectives of spatial spillover and time lag to determine the dynamic effects of TLT on economic growth and further discuss the heterogeneous characteristics of the economic growth across the different stages created by TLT.

The remainder of the paper is structured as follows: Section 2 summarizes the literature review and research hypotheses. Section 3 introduces the data source and empirical design. Section 4 reports the empirical results. Section 5 provides a more extensive discussion, and Section 6 presents the conclusion and policy insights.

## 2. Literature Review and Research Hypotheses

Land is an essential resource and is generally recognized by economists to play a significant role in economic growth [15–19]. As an important type of land resource, transportation land significantly impacts economic activities [20,21]. We analyze the effect of the TLT transmission process on economic growth from three aspects: intermediary effect, time-space lag transmission effect, and stage heterogeneity effect. This yields the research hypotheses.

### 2.1. Effects of TLT on Employment Promotion, Industrial Interaction, and Efficiency Improvement

Effective development and construction of transportation land can attract capital inflow, create employment opportunities, and shape the vitality of the regional economy [22–24]. The construction and operation of urban transportation land require large quantities of labor, material resources, and services. These can quickly facilitate the development of industry, transportation, commerce, and other relevant industries [25–27], stimulate employment [28,29], and increase the value of surrounding land [30]. This yields significant economic externalities; that is, the social and economic benefits generated by the project exceed the book profits of the project. While the profit potential of urban transportation land projects is limited, the economic impact can be amplified. As society develops, the population increases, and road network expansions, enhanced service levels, and urban transportation attract greater passenger flows. In the long run, the resulting operating income experiences stable growth, and its affiliated operating assets, such as advertising and shops, also appreciate. While driving employment growth, there is also a high development and construction demand for industrial products, building materials, living services, and productive services. Furthermore, the geographical and spatial characteristics of transportation land promote the vigorous development of local industries and the integration between industries. This is particularly the case with the deep integration of manufacturing and service industries, which effectively improves the efficiency of labor production [31].

With the advent of the new industrial revolution, there have been gradual improvements in the construction of new infrastructure, such as inter-city high-speed railways, urban rail transit, new energy charging piles, big data centers, 5G base stations, and artificial intelligence. This provides an information-sharing operation platform, forcing the integrated development of industries through intermediate investments. Driven by techno-

logical innovation, the traditional transportation industry is realizing digital transformation, intelligent upgrades, and integrated innovation to support high-quality development. Through periodic development and construction, transportation land has created a more perfect transportation infrastructure and a more effective spatial distribution configuration. This resulted in more efficient transportation services and lower transportation costs and time requirements, which promotes the optimization of the layout of different industries, reorganizes production activities in different regions [32–34], accelerates the flow of production factors, and improves the total factor productivity [35]. However, the excessive supply and development of transportation land may generate a "crowding-out effect" that slows down economic growth [36]. Based on the literature review, we speculate that TLT may promote economic growth to a certain extent, leading to Hypothesis 1:

**Hypothesis 1 (H1).** *TLT plays a significant role in promoting economic growth.*

*2.2. Time–Space Lag Conduction Effect of TLT*

Transportation land is the main carrier of transportation infrastructure and determines the layout, type, and level of land supply at its source. This ensures the effective performance of the land-use function [37,38]. The network layout of transportation land leads to a more rapid circulation of passenger flow, logistics, capital flow, information flow, and other resources and services in the network within and between cities. This is due to the powerful aggregation and release effect, which changes modes of consumption, life, and production and has thereby profoundly affected the economic operation of the city [39,40].

Therefore, the transportation land network creates the attributes of an economic circle that scales, and the radial influence covers most areas within the network and surrounding areas. Different types of transportation land place a different emphasis on different functions. Urban road land, urban rail transit land, traffic station land, and other types of transportation land supply can effectively alleviate traffic congestion [41,42], promote population flow [43,44], and strengthen the connection between regions [45]. Railway, airport, port, wharf, pipeline transport, and other types of transportation land have spatial spillover effects [46], strengthening cargo spatial transmission capacity and intensifying industrial cooperation [47]. Transportation infrastructure functions need a periodic development and construction of transportation land, which has a time-lag effect [48]. In summary, we speculate that TLT may have spatial spillover and time-lag effects on economic growth, leading to research Hypothesis 2:

**Hypothesis 2 (H2).** *TLT has spatial spillover and time-lag effects on economic growth.*

*2.3. Phase Heterogeneity Effect of TLT*

TLT is influenced by social and economic development [48]. Industrialization and urbanization greatly affect the scale, structure, type, and spatial layout of TLT [49,50]. Transportation land is the foundation of social and economic development, and land and development are mutually conditional and interdependent. Economic growth naturally drives increased passenger transport and cargo transport volume, further increasing demand for transportation land [51]. There is a positive correlation between TLT and social and economic development, which increases with the growth of economic aggregation, industrialization, and urbanization [52,53]. The scale and type of demand for transportation land depend on the level of socioeconomic development and industrial structure. The levels of economic development and industrial structure are distinct in different stages of industrialization, with significant changes in the characteristics of cargo transport demand [52].

For example, in the early stage of industrialization, the scale of freight volume continues to expand and the growth accelerates. In the later stage of industrialization, freight volumes are large and the growth rate gradually declines. In the post-industrialization stage, the freight volume increases steadily but with a low growth rate. With industrial

structure upgrades, the demand for transportation land type also significantly changes [54]. For example, the service industry, which is oriented to passenger flow, tends to be concentrated in densely populated areas with clear demands for facilitating the high-speed movement of people inside the city. As such, there is an urgent need for rail transit and other types of land. Urbanization, which is a general trend of socioeconomic development, profoundly impacts the transformation of social structure and the realization of modernization [55]. The layout of the transportation network serves as a guide for urban spatial expansion and impacts the urban development pattern to meet the increasing demand for passenger transport created by rapid urbanization [56]. The demand for transportation land changes during different stages of urbanization, especially during the change from disorderly spread to order [57]. Cities generally urgently need rail transit land to meet demands for convenient and efficient travel [58], and urban high-speed rail is needed between cities to meet the demands of rapid cross-city flow [43]. Therefore, the demand and supply of transportation land differ in different development stages. In summary, we speculate that TLT has a stage-based heterogeneous impact on economic growth, leading to Hypothesis 3:

**Hypothesis 3 (H3).** *TLT has heterogeneous effects on economic growth with respect to the economic development stage, industrial structure, and urbanization level.*

### 3. Data Resources and Empirical Design

Firstly, the spatial econometric model and time-lag model are used to investigate the spatial and temporal lag effects of TLT on economic growth. Then, the mechanism of action is explored by the mediation effect model. Finally, through the threshold effect model, we analyze the stage heterogeneity of TLT on economic growth.

*3.1. Variable Selection*

The types of variables used in this study are further explained below. In addition, Table 1 lists all defined variables including their names, types, symbols, and calculation methods.

3.1.1. Explained Variable

This paper investigates the relationship between TLT and regional economic growth. Per capita gross domestic product (GDP), deflated to eliminate the factor of price with 2007 as the base period, is used to measure the regional economic level as the explained variable. The following economic variables are used in the study; all have been deflated.

3.1.2. Core Explanatory Variable

Data about the TLT area (the newly transferred transportation land per year) were sourced from the China Land Market Network [59], which is the dynamic system that monitors the Chinese land market. The system integrates information release, monitoring and analysis, and sharing services, including data information, policies and regulations, and news about land supply in national and key areas. The China Land Market Network is the most comprehensive, timely, and accurate government website with information about China's land market. The TLT area serves as the core explanatory variable and measures the supply of transportation land in each province. According to the "Guidelines on the Classification of Land for Land Use for Territorial Space Investigation, Planning and Use Control", issued by the Ministry of Natural Resources in December 2020, transportation land generally includes the following nine types of land use: railway, highway, airport, port and wharf, pipeline transportation, urban rail transit, urban roads, transportation station, and other transportation facilities.

### 3.1.3. Other Variables

According to previous studies, control variables are defined that control other factors affecting economic growth [60,61]. Intermediate variables and threshold variables are also defined in Table 1.

**Table 1.** The explanations of all variables.

| Variable Types | Symbol | Variable Name | Calculation Method |
|---|---|---|---|
| explained variable | *pgdp* | regional economic level | per capita gross domestic product (GDP) |
| core explanatory variable | *land* | transportation land transfer (TLT) | transportation land transfer (TLT) area (the newly transferred transportation land per year) |
| control variable | *industry* | industrial structure | ratio of the added value of the tertiary industry to the added value of the secondary industry |
| control variable | *gover* | financial expenditure scale | proportion of local financial general budget expenditures to the total GDP |
| control variable | *edu* | education level | proportion of local financial education expenditure to the total GDP |
| control variable | *peo* | population size | number of permanent residents at the end of the year |
| control variable | *invest* | scale of foreign investment | ratio of total investment from foreign-invested enterprises to the total GDP |
| control variable | *trade* | trade scale | proportion of total import and export of goods to the total GDP |
| control variable | *elder* | degree of population aging | ratio of the population aged 65 and over to the population aged 15–64 |
| control variable | *tfp* | total factor productivity | stochastic frontier approach (SFA) |
| intermediate variable | *employment* | level of employment | number of urban unit employees |
| intermediate variable | *industry2* | interactive development of industries | added value of secondary and tertiary industries |
| intermediate variable | *efficiency* | overall efficiency of the economy | total factor productivity |
| threshold variable | *ur* | urbanization rate | proportion of urban population to the total population |
| threshold variable | *ind-str* | industrial structure | added value of the proportion of tertiary industry to total GDP |
| threshold variable | *ad-ind-str* | advanced industrial structure | two ratios were calculated to determine the index of advanced industrial structure. See text for details. |

### 3.2. Data Resources

Based on data availability, 30 provinces and autonomous regions in China serve as the research area for this study. The data do not include the Tibet Autonomous Region, Taiwan Province, Hong Kong Special Administrative Region, and Macao Special Administrative Region. The time span is from 2007 to 2019. As noted previously, data about the TLT area are from the China Land Market Network [60]. Data for the explained variable and other control variables are from the National Bureau of Statistics and provincial statistical yearbooks. The data about the industrial structure are obtained from the China Industrial Enterprises Statistical Yearbook, the China High-tech Industry Statistical Yearbook, and the China Science and Technology Statistical Yearbook.

### 3.3. Model Specifications

#### 3.3.1. Spatial Econometric Model

Spatial autocorrelation analysis: According to the literature review, the development and construction of transportation land may play a direct or indirect role in driving regional economies [35,62]. Therefore, this paper hypothesizes that the TLT has spatial spillover effects on economic growth. However, before hypothesis testing, a spatial autocorrelation test had to be conducted to determine whether the spatial econometric model can be used

for analysis. For this test, Moran's I index of transportation land transfer from 2007 to 2019 was calculated using an adjacency weight matrix. The result showed that Moran's I index values were all positive and highly significant between 0 and 1, indicating the occurrence of significant spatial agglomeration effects in the distribution of China's TLT.

Lagrange multiplier (LM) test: An LM test was applied to assess the spatial lag term and spatial error term. All test results of the spatial weight matrix rejected the null hypothesis that the equation has no spatial lag term and error term at a 1% significance level. Therefore, the equation contains both a spatial lag term and spatial error term. After a comparative analysis and based on past studies [63,64], the spatial Durbin model (SDM) was applied to analyze the spatial spillover effect of TLT on economic growth.

Hausman test and likelihood ratio (LR) test: This study used inter-provincial data for analysis. Each province has its own different economic and social conditions, pointing to the applicability of the fixed-effects model. The Hausman test and LR test were also applied. Based on past studies [65], the bidirectional fixed effects model, including space and time, was used for estimation.

Wald test and LR test: To select the model, we referred to Elhorst, Hui, and Liang [66,67]. After the SDM was estimated, the Wald test and LR test were conducted. All results rejected the null hypothesis, that is, the SDM could not be degraded into a spatial error model or a spatial autoregression model. Therefore, the SDM, which generalizes both the spatial error model and spatial autoregression model, is suitable for this research. As such, the SDM was used for the following analysis.

Based on the above assumptions and tests and considering the possible spatial spillover effect between individuals, we established a spatial Durbin panel regression model to assess the impact of TLT on economic growth:

$$ln(pgdp)_{it} = a_0 + b_1 \sum_{j=1}^{n} w_{ij} ln(pgdp)_{jt} + a_1 ln(land)_{it} + b_2 \sum_{j=1}^{n} w_{ij} ln(land)_{jt} + d \sum X_{it} + l \sum_{j=1}^{n} w_{ij} X_{jt} + n_i + m_i + e_{it} \quad (1)$$

where $i$ represents the province, $t$ represents the year, $v_i$ is the non-observable individual heterogeneity that does not change over time, $\mu_t$ is the non-observable factor that changes over time, and $\varepsilon_{it}$ is the random disturbance term. The parameters $\alpha, \beta, \delta, \lambda$ are a series of coefficients to be estimated. The spatial proximity weight matrix and spatial distance weight matrix ($w_{ij} = 1/d_{ij}$) are constructed according to the adjacency, distance, and positions of provinces; when two provinces are adjacent, $w_{ij} = 1$, otherwise the value is 0. The variable $land_{it}$ is the index of the TLT area, and $land_{jt}$ is the index of the TLT area in other provinces. The variable $pgdp_{it}$ is the index of the economic growth, and $pgdp_{jt}$ is the index of the economic growth in other provinces, measured by the per capita GDP. The variable $X_{it}$ is the set of control variables, and $X_{jt}$ is the set of control variables in other provinces.

### 3.3.2. Mediation Effect Model

After establishing the econometric model of the factors influencing economic growth, a mediation effect model is introduced to further investigate the direct impact of TLT on economic growth and the indirect impact of TLT on economic growth through employment, industrial interaction, and economic operation efficiency. A mediation effect model shows that when the influence of explanatory variable X on explained variable Y is decomposed, there is a direct influence of X on Y and an indirect influence on Y through the intermediate variable M. Therefore, M is the intermediate variable. In other words, the intermediate variable is the internal medium through which the explanatory variable indirectly affects the explained variable.

The mechanism reflected through the mediation effect is consistent with Hypothesis 2. Therefore, Hypothesis 2 is tested using mediation effect analysis. Three steps are used to test the mediation effect. First, regression Equation (1) of explanatory variable X to explained variable Y is constructed to test whether the coefficient of X is significant. If it is

not significant, there is no stable relationship between the two variables, and there is no mediation effect. If the regression coefficient is significant, the second step test is conducted, which involves constructing regression Equation (2) assessing the relationship between explanatory variable X and intermediate variable M. The third step involves constructing Equation (3), assessing the relationship of explanatory variable X and intermediate variable M to explained variable Y, and testing the significance of the regression coefficient again. If all three tests pass, the results support the presence of a mediation effect.

In the mediation effect model of this study, X is the TLT, M represents employment, industrial added value, and total factor productivity, and Y is the economic growth. Using employment as an example, $f_1, f_2, f_3$ are the total effects of X on Y, $(f_1, f_2, f_3) \times \varphi_4$ are the mediation effect transmitted through the intermediate variable M, and $\varphi_1, \varphi_2, \varphi_3$ is the direct effect of X on Y. The specific relationship is shown in Figure 1. The following measurement test model is constructed based on the analysis above:

$$ln(pgdp)_{it} = f_0 + f_1 ln(land)_{it} + f_2 ln(land)_{it-1} + f_3 ln(land)_{it-2} + f_4 \sum X_{it} + d_{it} \tag{2}$$

$$ln(employment)_{it} = j_0 + j_1 ln(land)_{it} + j_2 ln(land)_{it-1} + j_3 ln(land)_{it-2} + j_4 \sum X_{it} + e_{it} \tag{3}$$

$$ln(industry)_{it} = h_0 + h_1 ln(land)_{it} + h_2 ln(land)_{it-1} + h_3 ln(land)_{it-2} + h_4 \sum X_{it} + g_{it} \tag{4}$$

$$ln(tfp)_{it} = x_0 + x_1 ln(land)_{it} + x_2 ln(land)_{it-1} + x_3 ln(land)_{it-2} + x_4 \sum X_{it} + w_{it} \tag{5}$$

$$ln(pgdp)_{it} = \varphi_0 + \varphi_1 ln(land)_{it} + \varphi_2 ln(land)_{it-1} + \varphi_3 ln(land)_{it-2} + \varphi_4 ln(employment)_{it} + \varphi_5 \sum X_{it} + \mu_{it} \tag{6}$$

$$ln(pgdp)_{it} = s_0 + s_1 ln(land)_{it} + s_2 ln(land)_{it-1} + s_3 ln(land)_{it-2} + s_4 ln(industry)_{it} + s_5 \sum X_{it} + \rho_{it} \tag{7}$$

$$ln(pgdp)_{it} = \psi_0 + \psi_1 ln(land)_{it} + \psi_2 ln(land)_{it-1} + \psi_3 ln(land)_{it-2} + \psi_4 ln(tfp)_{it} + \psi_5 \sum X_{it} + \theta_{it} \tag{8}$$

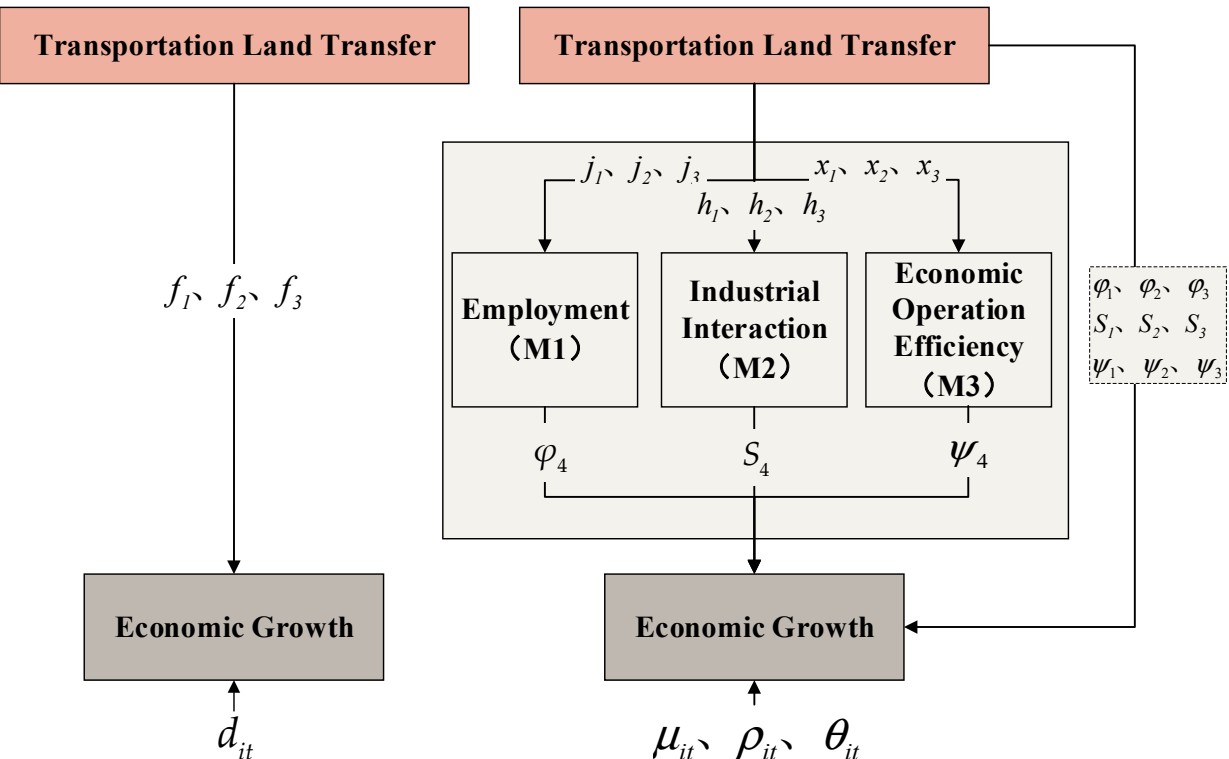

**Figure 1.** Path associated with the mediation effect theory. Notes: Transportation Land Transfer corresponds to $ln(land)_{it}, ln(land)_{it-1}, ln(land)_{it-2}$ in Equations (2)–(8); Economic Growth corresponds to $ln(pgdp)_{it}$ in Equations (2)–(8).

## 4. Results

Before analyzing the empirical results, it is useful to accurately assess the data to understand the supply of transportation land in each province. Figure 2 shows the distribution of the TLT area in the 30 provinces of China from 2007 to 2019. In the early stage, the TLT area was concentrated in the eastern region. It gradually moved to the central and western regions during the middle and late periods in a wave pattern. The supply of transportation land is highly correlated with the stage of economic development and depends on the demand for transportation, based on the economic stage and industrial structure at that time. The change in the degree of TLT concentration is sequentially connected in the eastern and western regions of China, and there is a clear spatial conduction effect. For areas experiencing a middle and lower stage of economic development, TLT reflects quantitative expansion. For areas in a higher development stage, the effect of quantitative expansion is weakened, and there is a stronger qualitative promotion effect to promote economic growth. As a result, large-scale development of new land is not required. Therefore, in this stage, it is difficult to accurately measure the effect of TLT on economic growth.

Figure 3 shows the density of the TLT area in the 30 provinces from 2007 to 2019, represented by the ratio of the TLT area to the total area of the provinces. The results show a significant spatial difference in the density of TLT; the temporal trend is less clear. The density of TLT shows a three-stage decreasing trend in the eastern, central, and western regions.

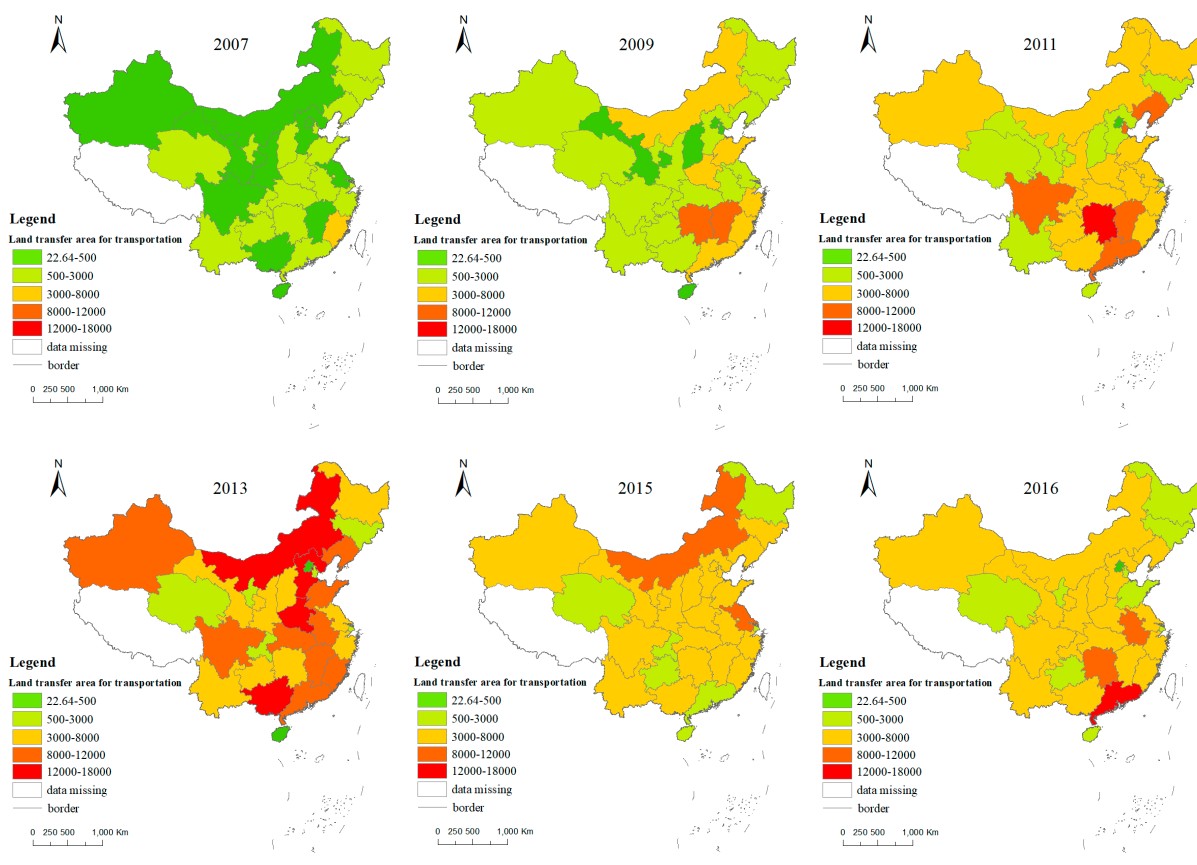

**Figure 2.** *Cont.*

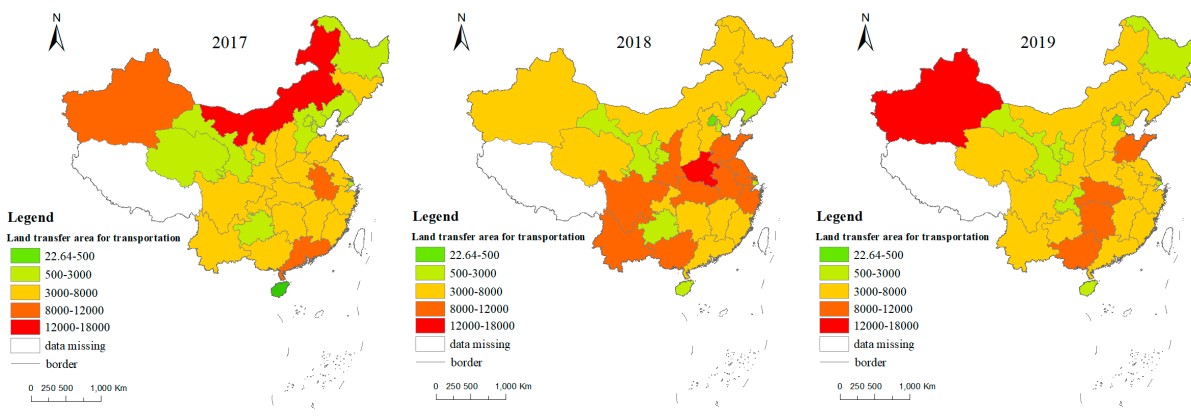

**Figure 2.** Spatial and temporal distribution and evolution trend of the TLT area.

**Figure 3.** Spatial and temporal distribution and evolutionary trend of TLT density.

### 4.1. Analysis of the Spatial Spillover Effects and Time-Lag Effects of TLT on Economic Growth

This paper applied the direct and indirect utility theory proposed by LeSage and Pace [68], which explains the model variable parameters as "cumulative effects," composed of direct and indirect effects (spatial spillover effects). This theory avoids the problem of imprecision when using a spatial lag variable coefficient to assess a spatial spillover effect.

The regression results assessing TLT against economic growth are listed in Table 2, columns 1–2. To test the robustness of the empirical results, this paper applied the methods of increasing control variables, changing the spatial weight matrix, and changing the econometric model. In Table 2, columns 1–3 contain the regression results generated by using the spatial adjacency matrix, and columns 4–6 list the regression results generated by using the spatial distance weight matrix. The results in Table 2 inform the following conclusions.

**Table 2.** Direct effect, indirect effect, and total effect of the spatial Durbin model.

| Column | 1 | 2 | 3 | 4 | 5 | 6 |
|---|---|---|---|---|---|---|
| Model Form | Spatial Adjacency Weight Matrix | | | Spatial Distance Weight Matrix | | |
| Type | SDM_FE | | | SDM_FE | | |
| | Direct | Indirect | Total | Direct | Indirect | Total |
| Variables | ln(pgdp) | ln(pgdp) | ln(pgdp) | ln(pgdp) | ln(pgdp) | ln(pgdp) |
| ln(land) | 0.0377 *** | 0.1249 ** | 0.1626 *** | 0.0115 ** | 0.0707 * | 0.0823 * |
| | (3.1795) | (2.3882) | (3.0230) | (2.3595) | (1.7062) | (1.9433) |
| ln(industry) | 0.1098 | 0.5965 *** | 0.7063 ** | −0.1542 *** | 0.0628 | −0.0914 |
| | (0.8999) | (2.7153) | (2.4649) | (−4.6929) | (0.4023) | (−0.5539) |
| ln(govern) | 0.1291 | −0.0057 | 0.1234 | −0.1596 ** | −0.4209 | −0.5805 |
| | (1.3245) | (−0.0124) | (0.2427) | (−2.4706) | (−0.8967) | (−1.2561) |
| ln(edu) | −0.0122 | 0.0314 | 0.0192 | 0.0031 | 0.1139 | 0.1170 |
| | (−0.4098) | (0.3163) | (0.1648) | (0.1452) | (0.7022) | (0.6868) |
| ln(peo) | 0.8652 * | 2.3345 | 3.1997 ** | 0.0975 | 1.4435 | 1.5410 |
| | (1.8378) | (1.5823) | (2.4319) | (0.4099) | (1.1738) | (1.1428) |
| ln(invest) | −0.0998 ** | −0.0093 | −0.1090 | −0.0133 | −0.0564 | −0.0698 |
| | (−2.4948) | (−0.0472) | (−0.4946) | (−0.9195) | (−0.5078) | (−0.5891) |
| ln(trade) | 0.0128 | 0.2356 | 0.2484 | 0.0089 | −0.0147 | −0.0058 |
| | (0.3311) | (1.3319) | (1.3841) | (0.2517) | (−0.0878) | (−0.0327) |
| ln(elder) | 0.0639 | 0.2327 | 0.2965 | −0.0499 | −0.0607 | −0.1105 |
| | (1.0032) | (0.9382) | (1.1428) | (−1.2053) | (−0.3461) | (−0.6363) |
| ln(tfp) | 0.0113 | 0.0066 | 0.0179 | 0.0099 | −0.0129 | −0.0030 |
| | (1.4125) | (0.4480) | (1.1939) | (1.3233) | (−0.7889) | (−0.1709) |
| Year FE | yes | yes | yes | yes | yes | yes |
| Province FE | yes | yes | yes | yes | yes | yes |
| Observations | 390 | 390 | 390 | 390 | 390 | 390 |
| $R^2$ | 0.0198 | 0.0198 | 0.0198 | 0.0006 | 0.0006 | 0.0006 |
| Numbers of provinces | 30 | 30 | 30 | 30 | 30 | 30 |

Notes: *z* statistics in parentheses; ***, **, and * indicate significance at the 1%, 5% and 10% levels; *ln(variable)* refers to the logarithm of this variable.

TLT has spatial spillover effects on economic growth. After applying different weight matrices, the regression coefficients of *ln(land)* are all positive. The regression coefficients of *ln(land)* are also positive when different models are established. This indicates that TLT has significant driving effects on economic growth in each province and significant spatial spillover effects. Using the spatial adjacency matrix as an example, the regression results in columns 1–3 of Table 2 show that for every 1% increase in *ln(land)*, the direct and indirect effects (spatial spillover effects) promote the increase of *ln(pgdp)* by 0.0377% and 0.1249%, respectively, with a cumulative effect of 0.1626%.

As a possible reason for the spatial spillover effect, the transport infrastructure construction after TLT will not only promote local employment but also employment in

adjacent regions. In addition, the efficiency of local economic operation will be improved through the increased supply of transportation land. As a result, the close combination of economic activities in one region is easily realized with its adjacent regions.

TLT has time-lag effects in promoting economic growth. The regression results in columns 7–8 of Table 3 show that regardless of whether the effect is fixed or random, the regression coefficients of *ln(land)* are positive at a 1% significance level. This confirms the robustness of the empirical results. The TLT itself does not have significant economic promotion effects. However, the consequent development and construction of transportation land and the subsequent infrastructure operations play a real role in promoting economic growth when a time-lag term for the TLT area is added based on the original model.

**Table 3.** Time-lag effect of TLT on economic growth.

| Column | 7 | 8 | 9 | 10 |
|---|---|---|---|---|
| Model Form | Ordinary Panel Models | | Time-Lag Panel Model | |
| Type | FE | RE | First-Order Lag | Second-Order Lag |
| Variables | *ln(pgdp)* | *ln(pgdp)* | *ln(pgdp)* | *ln(pgdp)* |
| *ln(land)* | 0.0082 ** (2.3445) | 0.0983 *** (3.8709) | 0.0050 (1.4071) | 0.0018 (0.5141) |
| *l1.ln(land)* | - | - | 0.0076 ** (2.2958) | 0.0063 * (1.9550) |
| *l2.ln(land)* | - | - | - | 0.0027 (0.8782) |
| *ln(industry)* | −0.1731 *** (−7.9752) | 0.6080 *** (5.1773) | −0.1484 *** (−6.8149) | −0.1209 *** (−5.6461) |
| *ln(gover)* | −0.1221 *** (−3.2425) | 0.6511 *** (5.0752) | −0.1486 *** (−4.1630) | −0.2345 *** (−5.9185) |
| *ln(edu)* | 0.0045 (0.2129) | 0.0202 (0.2150) | 0.0089 (0.4501) | 0.1274 *** (3.4602) |
| *ln(peo)* | 0.0729 (1.0032) | 0.3576 *** (3.8231) | 0.1740 ** (2.2536) | 0.2843 *** (3.5893) |
| *ln(invest)* | −0.0154 (−1.5198) | 0.1383 (1.5307) | −0.0161 (−1.5972) | −0.0125 (−1.2429) |
| *ln(trade)* | 0.0386 *** (3.6258) | 0.1084 * (1.8369) | 0.0366 *** (3.4171) | 0.0306 *** (2.9393) |
| *ln(elder)* | −0.0568 * (−1.7216) | 0.1645 (1.1630) | −0.0497 (−1.5780) | −0.0470 (−1.5976) |
| *ln(tfp)* | 0.0098 (1.3669) | 0.0468 *** (3.9373) | 0.0086 (1.2849) | 0.0030 (0.3962) |
| constant | 8.8160 *** (13.4765) | 6.2629 *** (7.6362) | 8.0546 *** (11.6846) | 7.6132 *** (10.8635) |
| year FE | yes | no | yes | yes |
| province FE | yes | no | yes | yes |
| observations | 390 | 390 | 390 | 390 |
| F-value/chi$^2$ | 1113.1271 | 438.1977 | 1002.8024 | 912.8071 |
| R$^2$ | 0.3356 | 0.3856 | 0.2531 | 0.1529 |
| number of provinces | 30 | 30 | 30 | 30 |

Notes: *t* statistics in parentheses; ***, **, and * indicate significance at the 1%, 5% and 10%, levels; *ln(variable)* refers to the logarithm of this variable.

To determine the lag period, this paper referred to the "Idle Land Disposal Method," the "Land Management Law," and the "Urban Real Estate Management Law." In China, the government can collect idle-land fees from enterprises that have not started development within one year. Furthermore, the government can reclaim the right to use state-owned construction land free of charge, if the enterprise has not started construction within two years. Therefore, under normal circumstances, enterprises will develop and construct transportation land within one or two years. As such, we address TLT with one-stage

and two-stage lags. The results in columns 9–10 of Table 3 show that the coefficient of the first-stage lag is significantly positive, while the coefficient of the second-stage lag is positive but not significant. This further supports the time-lag effect of TLT on economic growth.

The regression coefficients of the control variables are essential on the same levels as predicted. The industrial structure, the scale of fiscal expenditures, the educational level, the scale of population, the scale of foreign trade, and total factor productivity are positively correlated with economic growth. This may relate to improvements in China's industrial structure, efficient fiscal expenditures, effective educational structure, sufficient human resources, increased opening to the outside world, the acceleration of scientific and technological progress, and other factors. The scale of foreign investment is negatively correlated with economic growth. This may be related to a high crowding-out effect and low quality of the imported foreign investment. The dependency ratio of the elderly population is positively correlated with economic growth, which might be attributed to the development of medical and health care industries, created by the aging of the population.

The policy implications of these empirical results are as follows. To encourage the economic and social development of each province, the authorities should consider the overall economic situation, strengthen the interaction of regional traffic, and rationally allocate land resources. Appropriately expanding the scale of transportation land can effectively improve the supporting industrial level, optimize the layout of urban functions, and strengthen economic growth.

### 4.2. Analysis of the Mediation Effects of TLT on Economic Growth

The analysis above indicates that TLT has spillover effects in space and lag effects in time on economic growth; however, the path of action needs further exploration. Although TLT has time-lag effects on economic growth, economic growth is directly promoted by the subsequent construction of transportation facilities [69]. First, development and construction can increase employment in transportation, construction, manufacturing, accommodation, and catering industries [70,71]. Second, development and construction can drive the interactive development of industries [72]. Finally, the spillover effects of TLT on economic growth can also be explained by the fact that transportation land may improve the overall efficiency of the economy (characterized by the total factor productivity, tfp) by improving the transportation network and indirectly encouraging economic growth.

The direct impact of TLT on economic growth and the intermediate effect generated through employment, industrial interaction, and economic efficiency was assessed using Equations (2)–(8). The specific regression analysis results are listed in Table 4. There are time-lag effects between TLT and the actual impacts. Consistent with the analysis above, the first-order and second-order lag terms of transportation land are introduced into the model. Column 11 in Table 4 presents the overall effect of the first-order and second-order lags of TLT on economic growth. This indicates that the coefficient of the first-order lag term of TLT on economic growth is 0.0063, which is significantly positive at a 5% significance level. The TLT itself has very limited effects on economic growth, verified by the significant lag coefficient. In fact, some transportation land supply is provided by the government for free.

TLT has significantly positive impacts on economic growth; as such, the mediation effect test continues. Columns 12–14 in Table 4 show the influences of TLT on intermediate variables (number of urban unit employees, added value of secondary and tertiary industries, total factor productivity). The regression coefficients of the lag term are significantly positive. In columns 15–17 of Table 4, the regression coefficients of urban unit employment, the added value of the secondary and tertiary industries, and total factor productivity are all significantly positive, indicating the presence of mediation effects. TLT directly impacts economic growth and has a mediation effect through employment, industrial interactions, and economic efficiency.

In addition, the impact of urban unit employment is assessed in the subsequent tests. The results indicate that the lag effect of TLT can significantly promote an increase in employment in transportation, warehousing and postal services, construction, manufacturing, accommodation, and catering industries. This increased employment plays a mediation effect in promoting economic growth. Further testing of the industrial interactions reveals similar findings. When examining the added value of the secondary and tertiary industries, we also find the presence of mediation effects on promoting economic growth. In addition, after the deep test of the structural equation model, we obtained a more robust conclusion about the mediation effects, as shown in Table 5. Therefore, through the analysis of the mediation effect, we obtained the mechanism by which TLT promotes economic growth, and the action path is shown in Figure 4.

**Table 4.** Mediation effects of TLT on economic growth.

| Column | 11 | 12 | 13 | 14 | 15 | 16 | 17 |
|---|---|---|---|---|---|---|---|
| **Model Form** | **First** | | **Second** | | | **Third** | |
| **Type** | | **Employment** | **Industry** | **Efficiency** | | | |
| **Variables** | *ln(pgdp)* | *ln(emp)* | *ln(ind)* | *ln(tfp)* | *ln(pgdp)* | *ln(pgdp)* | *ln(pgdp)* |
| *ln(land)* | 0.0017 (0.4952) | −0.0000 (−0.0059) | 0.0089 (1.0285) | −0.0208 (−0.7631) | 0.0017 (0.4835) | −0.0011 (−0.4646) | 0.0005 (0.1540) |
| *l1.ln(land)* | 0.0063 ** (1.9721) | 0.0065 (0.9366) | 0.0162 ** (1.9836) | 0.0118 (0.4567) | 0.0057 * (1.7023) | 0.0020 (0.8563) | 0.0004 (0.1449) |
| *l2.ln(land)* | 0.0030 (0.9846) | 0.0126 * (1.9494) | 0.0097 (1.2433) | 0.0862 *** (3.5793) | 0.0042 (1.3630) | −0.0001 (−0.0384) | 0.0023 (0.8433) |
| *ln(emp)* | - | - | - | - | 0.1427 *** (5.0480) | - | - |
| *ln(ind)* | - | - | - | - | - | 0.2760 *** (16.1227) | - |
| *ln(tfp)* | - | - | - | - | - | - | 0.0111 * (1.7046) |
| control | yes | yes | yes | yes | yes | yes | yes |
| year FE | yes | yes | yes | yes | yes | yes | yes |
| province FE | yes | yes | yes | yes | yes | yes | yes |
| observations | 330 | 330 | 330 | 330 | 330 | 330 | 330 |
| F-value | 961.3341 | 46.0773 | 209.1668 | 60.7261 | 1047.1166 | 1860.4953 | 897.2644 |
| R² | 0.9856 | 0.7463 | 0.9373 | 0.8042 | 0.9844 | 0.9929 | 0.9889 |
| number of provinces | 30 | 30 | 30 | 30 | 30 | 30 | 30 |

Notes: *z* statistics in parentheses; ***, **, and * indicate significance at the 1%, 5% and 10% levels; *ln(variable)* refers to the logarithm of this variable.

The conclusions above are universal and general; however, they lack specific applicability and relevant distinctions, considering that employment, industry, and economic efficiency may differ in different provinces given the different stages of economic development in those provinces. Therefore, to study the heterogeneous effects of regional TLT on economic growth at different stages, further econometric analysis was conducted by distinguishing different stages of economic development.

### 4.3. Analysis of the Effect of Stage Heterogeneity of TLT on Economic Growth

The previous theoretical analysis confirmed that TLT has a positive impact on regional economic growth. However, the impact on economic growth may differ in different economic development stages. As such, this paper applies Chenery's classification criteria for economic development stages, as shown in Table 6. Comprehensively considering the average per capita GDP of each province and the proportion of the per capita GDP of each year in the classification range, we categorize the development stages of the 30 studied provinces. Table 7 lists the classification results. To test whether TLT has different

impacts on economic growth at different stages of economic development, we conducted a regression analysis of samples for regions at different stages.

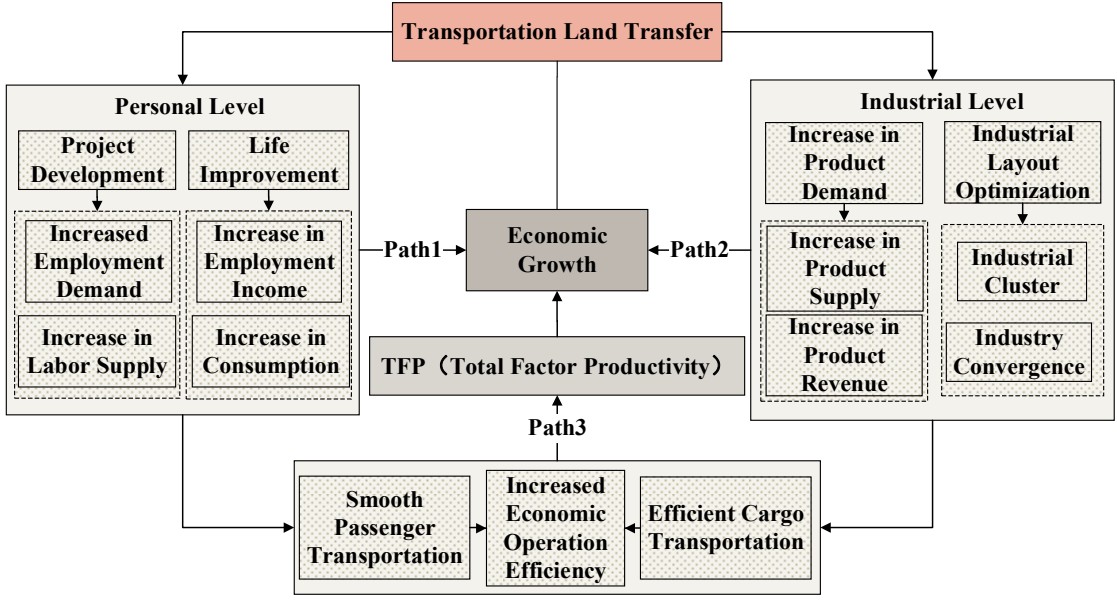

**Figure 4.** Mediation effect action path.

**Table 5.** Results of the structural equation model.

| Column | 18 | 19 | 20 | 21 | 22 | 23 |
|---|---|---|---|---|---|---|
| **Type** | **Employment** | **Industry** | **tfp** | **Employment** | **Industry** | **tfp** |
| **Variables** | *ln(Employment)* | *ln(Industry)* | *ln(tfp)* | *ln(pgdp)* | *ln(pgdp)* | *ln(pgdp)* |
| *ln(land)* | 0.2351 *** (6.9434) | 0.3777 *** (10.3798) | 0.1377 *** (5.1362) | −0.0661 *** (−2.6848) | −0.0955 *** (−4.5587) | 0.0491 ** (1.9692) |
| *ln(employment)* | - | - | - | 0.3524 *** (9.7867) | - | - |
| *ln(industry)* | - | - | - | - | 0.4388 *** (17.0096) | - |
| *ln(tfp)* | - | - | - | - | - | 0.1540 *** (3.3792) |
| *cons* | 4.1345 *** (15.0130) | 6.4575 *** (22.1022) | −0.8410 *** (−3.9080) | 8.8893 *** (37.0521) | 7.0400 *** (31.5133) | 10.0031 *** (50.6772) |

| estimates | type | Delta | | Sobel | | Monte Carlo | |
|---|---|---|---|---|---|---|---|
| | *ln(employment)* | 0.083 *** (5.663) | | 0.083 *** (5.663) | | 0.083 *** (5.494) | |
| indirect effect | *ln(industry)* | 0.166 *** (8.860) | | 0.166 *** (8.860) | | 0.167 *** (8.538) | |
| | *ln(tfp)* | 0.021 *** (2.823) | | 0.021 *** (2.823) | | 0.021 *** (2.797) | |

| | | | | | | |
|---|---|---|---|---|---|---|
| log(likelihood) | −1201.3942 | −1332.6993 | −1316.0624 | −1201.3942 | −1332.6993 | −1316.0624 |
| observations | 360 | 390 | 390 | 360 | 390 | 390 |
| number of provinces | 30 | 30 | 30 | 30 | 30 | 30 |

Notes: *z* statistics in parentheses; *** and ** indicate significance at the 1% and 5% levels; *ln(variable)* refers to the logarithm of this variable.

The regression results are shown in Table 8. Samples at all development stages show that TLT has significant promoting effects on economic growth. The promoting effect decreases as the development stages improve. This result also motivated the subsequent threshold regression. As the economic development stage improves and the economic

aggregate increases, the direct promoting effect of TLT on economic growth is diluted. This implies that other industries create more indirect effects. For example, the construction of transportation infrastructure improves the overall operational efficiency of the economy, creating implicit economic growth. This is supported by the analysis of the mediation effect above. Hence, the promotion effects of TLT on economic growth should not be underestimated simply due to the small regression coefficient of a high economic development stage. In contrast, for regions in a high economic development stage, transportation land needs to be scientifically and rationally planned, based on the overall economic situation. For regions in a low economic development stage, transportation land should not be blindly and disorderly expanded, simply due to the large promoting effect of TLT on economic growth.

**Table 6.** Classification criterion of economic development stages by Chenery.

| Stage | Stage Name | 1970 | 1990 | 1995 | 2000 | 2005 | 2010 |
|---|---|---|---|---|---|---|---|
| 1st stage | primary product stage I | 100–150 | 350–470 | 390–550 | 440–620 | 500–700 | 560–790 |
| | primary product stage II | 150–280 | 470–950 | 550–1100 | 620–1240 | 700–1410 | 790–1580 |
| 2nd stage | primary industrialized stage | 280–570 | 950–1890 | 1100–2210 | 1240–2480 | 1410–2830 | 1580–3150 |
| | middle industrialized stage | 570–1130 | 1890–3780 | 2210–4410 | 2480–4970 | 2830–5650 | 3150–6310 |
| | late industrialized stage | 1130–2100 | 3780–7070 | 4410–8250 | 4970–9330 | 5650–10,570 | 6310–11,820 |
| 3rd stage | primary developed stage | 2100–3360 | 7070–11,310 | 8250–13,200 | 9330–14,910 | 10,570–16,920 | 11,820–18,900 |
| | developed stage | 3360–5050 | 11,310–16,980 | 13,200–19,810 | 14,910–22,390 | 16,920–25,390 | 18,900–28,360 |

Notes: Table data represent annual GDP per capita in US dollars. The degree of precision is 10, which is consistent with World Bank data.

**Table 7.** China's 30 provinces, grouped by their stage of economic development.

| Stage Name | Province Name |
|---|---|
| primary industrialized stage | Guizhou, Gansu |
| middle industrialized stage | Hebei, Jilin, Heilongjiang, Shanxi, Anhui, Jiangxi, Henan, Hunan, Guangxi, Hainan, Sichuan, Yunnan, Shaanxi, Qinghai, Ningxia, Xinjiang |
| late industrialized stage | Neimenggu, Liaoning, Fujian, Shandong, Hubei, Guangdong, Chongqing |
| primary developed stage | Beijing, Tianjin, Shanghai, Jiangsu, Zhejiang |

*4.4. Analysis of the Threshold Effect of TLT Promoting Economic Growth*

TLT has a positive impact on economic growth, with actual results aligning with theoretical expectations. However, the regression coefficient is on an average level with respect to the economic development stage, urbanization level, and industrial structure. Thus, it is difficult to offer effective reference values for countries or regions with different development conditions. Therefore, we assess whether the effects of urbanization rate, industrial structure, and advanced industrial structure on economic growth possess nonlinear characteristics at different levels. This provides more comprehensive theoretical reference values for other countries or regions.

4.4.1. Threshold Effect Analysis of Urbanization Rate on TLT to Promote Economic Growth

The urbanization rate is the proportion of the urban population to the total population and an important indicator affecting economic growth [73]. The effects of TLT on economic

growth may vary in different urbanization rate intervals. To assess this assumption, the urbanization rate was set as the threshold variable to analyze the threshold effects.

The form of the panel threshold model was tested before estimating the model. To confirm the value and quantity of the threshold, a bootstrap sampling method was applied to simulate the likelihood ratio statistics 2000 times and thereby estimate the related statistics. The specific results are listed in Table 9. These estimation results indicate that the F-statistics of the single threshold and double threshold are significant at the 1% level, but the F-statistics of the triple threshold are not. Therefore, a double-threshold effect may exist in the urbanization rate, and the urbanization rate can serve as the threshold variable to investigate the influence of TLT on economic growth. The impact of TLT on economic growth will differ in different threshold intervals. Therefore, according to the different threshold values of urbanization rate, this paper sets dummy variables and generates cross-multiplier terms with TLT variables. This approach enables an investigation of the nonlinear relationship between TLT and economic growth. The specific form of the threshold model is set as follows:

$$
\begin{aligned}
ln(pgdp)_{it} = \ & \alpha_o + \alpha_1 ln(industry)_{it} + \alpha_2 ln(govern)_{it} + \alpha_3 ln(edu)_{it} + \alpha_5 ln(peo)_{it} + \alpha_6 ln(invest)_{it} + \alpha_7 ln(trade)_{it} + \alpha_8 ln(elder)_{it} \\
& + \alpha_9 ln(tfp)_{it} + \beta_1 ln(land)_{it} \cdot I(ur_{it} \leq \gamma_1) + \beta_2 lnland_{it} \cdot I(\gamma_1 < ur_{it} \leq \gamma_2) + \beta_3 ln(land)_{it} \cdot I(ur_{it} > \gamma_2) + v_{it}
\end{aligned}
\tag{9}
$$

**Table 8.** Phase-based heterogeneity of transportation land transfer to economic growth.

| Variables | Primary Industrialized Stage | Middle Industrialized Stage | Late Industrialized Stage | Primary Developed Stage |
|---|---|---|---|---|
| | *ln(pgdp)* | *ln(pgdp)* | *ln(pgdp)* | *ln(pgdp)* |
| *ln(land)* | 0.1088 * | 0.0841 *** | 0.0686 ** | 0.0320 *** |
| | (2.0545) | (5.5818) | (2.4600) | (3.7694) |
| *ln(industry)* | 0.1680 | 0.4966 *** | 0.6273 *** | 0.6382 *** |
| | (0.4225) | (6.2736) | (4.7797) | (9.3314) |
| *ln(govern)* | 0.1981 | 0.3755 ** | 1.1071 *** | −0.2324 * |
| | (0.5057) | (2.4978) | (4.6997) | (−1.7706) |
| *ln(edu)* | −0.0615 | 0.3491 ** | −0.5329 ** | 0.1241 |
| | (−0.7882) | (2.5068) | (−2.3322) | (0.8817) |
| *ln(peo)* | 5.2567 ** | 2.3531 *** | 2.6639 *** | 2.0327 *** |
| | (2.2779) | (6.6378) | (5.4419) | (6.8360) |
| *ln(invest)* | 0.2048 | −0.0071 | −0.3917 *** | −0.2538 *** |
| | (1.2069) | (−0.1683) | (−4.3172) | (−5.1229) |
| *ln(trade)* | −0.3222 ** | 0.0338 | −0.0039 | −0.2966 *** |
| | (−2.4755) | (0.7382) | (−0.0673) | (−4.0497) |
| *ln(elder)* | 0.3467 | 0.5027 *** | 0.4585 ** | 0.1102 * |
| | (0.3926) | (3.1168) | (2.2456) | (1.9933) |
| *ln(tfp)* | 0.0003 | 0.0273 | 0.0403 | 0.0105 |
| | (0.0068) | (1.4357) | (1.4308) | (1.2792) |
| *constant* | −32.0606 * | −9.3311 *** | −13.0380 *** | −2.4443 |
| | (−1.7835) | (−3.1114) | (−2.7727) | (−0.8072) |
| F | 36.2347 | 101.3088 | 69.5252 | 270.7611 |
| province FE | yes | yes | yes | yes |
| observations | 26 | 156 | 91 | 65 |
| $R^2$ | 0.9560 | 0.8710 | 0.8930 | 0.9795 |

Notes: *t* statistics in parentheses; ***, **, and * indicate significance at the 1%, 5%, and 10% levels; *ln(variable)* refers to the logarithm of this variable.

In Table 9, the threshold effect test results are divided into intervals according to the level of urbanization rate *ur*: interval 1 is *ur* ≤ 38.70%, interval 2 is 38.70% < *ur* ≤ 53.50%, and interval 3 is *ur* > 53.50%. Then, we analyzed the influence of TLT on economic growth using the urbanization rate in different intervals. The regression results of these three variables in different threshold intervals are given in Table 8. The estimation results of the coefficient of the control variables are consistent with the results above and are not

listed here. According to the regression results in Table 10, the regression coefficients of TLT on economic growth are all positive in the different threshold intervals. In other words, TLT has a consistent promoting effect on economic growth. As the urbanization rate crosses from the first to the second interval, there is a gradual increase in the promoting effect of TLT on economic growth. When crossing from the second to the third interval, the promoting effects of TLT on economic growth significantly increase. There are some possible reasons for this outcome.

**Table 9.** Test of the threshold effect of transportation land transfer on economic growth.

| Threshold Variable | Test Type | Threshold Value | F Value | p Value | 10% Critical Value | 5% Critical Value | 1% Critical Value |
|---|---|---|---|---|---|---|---|
| urbanization rate | single threshold test | 0.4353 *** | 76.99 | 0.0010 | 34.1310 | 40.4873 | 53.2495 |
| | double threshold test | 0.3870 *** 0.5350 *** | 46.12 | 0.0100 | 28.2019 | 33.9444 | 46.0744 |
| | three threshold test | 0.4469 | 72.59 | 0.5980 | 122.3648 | 137.5176 | 161.5293 |
| industrial structure | single threshold test | 0.4416 * | 27.15 | 0.0665 | 24.9042 | 29.8508 | 42.0312 |
| | double threshold test | 0.4416 0.7590 | 8.58 | 0.6245 | 21.7127 | 25.8614 | 37.0924 |
| | three threshold test | 0.5399 | 4.40 | 0.8260 | 17.6989 | 22.0327 | 31.7115 |
| advanced industrial structure | single threshold test | 0.0880 *** | 43.09 | 0.0125 | 26.0599 | 31.0918 | 44.5134 |
| | double threshold test | 0.1590 | 22.70 | 0.1300 | 25.4270 | 32.4303 | 44.2599 |
| | three threshold test | 0.0310 | 24.80 | 0.5510 | 47.0825 | 54.0334 | 68.2644 |

Notes: *t* statistics in parentheses; *** and * indicate significance at the 1% and 10% levels.

**Table 10.** Regression results showing the threshold effect of transportation land transfer on economic growth.

| Threshold Interval | Interactive Items | Coefficient Estimation |
|---|---|---|
| d1 (*advanced industrial structure* ≤ 8.80%) | lnland × d1 | 0.0792 *** (7.9485) |
| d2 (*advanced industrial structure* > 8.80%) | lnland × d2 | 0.0975 *** (9.5366) |
| d3 (*urbanization rate* ≤ 38.70%) | lnland × d3 | 0.0050 (0.4397) |
| d4 (38.70% < *urbanization rate* ≤ 53.50%) | lnland × d4 | 0.0495 *** (5.1890) |
| d5 (*urbanization rate* > 53.50%) | lnland × d5 | 0.0690 *** (7.5170) |
| d6 (*industrial structure* ≤ 44.16%) | lnland × d6 | 0.0717 *** (6.9252) |
| d7 (*industrial structure* > 44.16%) | lnland × d7 | 0.0859 *** (8.5094) |

Notes: *t* statistics in parentheses; *** indicate significance at the 1% levels.

First, areas with high urbanization rates are densely populated and economically active. This supports economies of scale for transportation land. As a large number of rural people migrate to cities, urban capacity has continuously expanded, urban construction

facilities have developed rapidly, comprehensive transportation hub systems have become more complete, and the implementation of high-speed rail, highways, subways, and other transportation facilities has improved. Hence, there are large opportunities for optimization and development. For areas with a high urbanization rate, continuing to expand TLT supports improved economies of scale. Second, during urbanization, the transfer of labor, capital, and other factors from inefficient to efficient sectors can be facilitated through effective supplies of transportation land, optimizing resource allocation, and improving production efficiency. Areas with high urbanization rates generally have a more favorable development environment, playing a critical role in attracting talent, capital, and other high-quality resources. In general, there are also highly efficient transportation land supplies in these areas, which supports the flow of factors through the means of transportation. This includes improving the flow path, reducing flow cost, and improving flow efficiency. Therefore, the total effect of TLT to promote economic growth in areas with high urbanization rates is generally higher compared to areas with low urbanization rates.

4.4.2. Threshold Effect Analysis of the Industrial Structure on TLT to Promote Economic Growth

The industrial structure is an important representation of the regional economic structure, represented by the added value of the proportion of tertiary industry to total GDP. The effects of TLT may vary due to the different industrial structure intervals. To verify this assumption, we set the industrial structure as the threshold variable and analyze the threshold effect.

After threshold model testing, threshold values and related statistics are estimated. Table 9 lists the specific results. The estimation results indicate a single threshold effect in the industrial structure. The impact of TLT on economic growth differs in the different threshold intervals. Therefore, given the different threshold values of the industrial structure, this paper establishes dummy variables and generates cross-multiplier terms with TLT variables to investigate the nonlinear relationship between TLT and economic growth. The specific form of the threshold model is as follows:

$$
\begin{aligned}
ln(pgdp)_{it} = \; & \alpha_o + \alpha_1 ln(industry)_{it} + \alpha_2 ln(govern)_{it} + \alpha_3 ln(edu)_{it} + \alpha_5 ln(peo)_{it} + \alpha_6 ln(invest)_{it} + \alpha_7 ln(trade)_{it} + \\
& \alpha_8 ln(elder)_{it} + \alpha_9 ln(tfp)_{it} + \beta_1 ln(land)_{it} \cdot I(ind-str_{it} \leq \delta_1) + \beta_2 ln(land)_{it} \cdot I(ind-str_{it} > \delta_1) + \nu_{it}
\end{aligned}
\tag{10}
$$

The threshold effect test results in Table 9 indicate that the industrial structure *ind-str* is divided into two intervals: interval 1 is *ind-str* ≤ 44.16%, and interval 2 is *ind-str* > 44.16%. The regression results in Table 10 indicate that in different threshold intervals, the regression coefficients of TLT on economic growth are all positive, that is, there is a consistent promoting effect of TLT on economic growth. As the industrial structure crosses from the first to the second stage, the promotion effect of TLT on economic growth experiences a stepwise reinforcement, possibly because the large-scale expansion of the service industry significantly improves the industrial structure. By maximizing the advantages of specialization, the service industry and other industries, especially the manufacturing industry, are encouraged to develop cooperatively and optimally. This strengthens the quality and efficiency of industrial development.

4.4.3. Threshold Effect Analysis of the Advanced Industrial Structure on TLT to Promote Economic Growth

Two ratios were calculated to determine the index of the advanced industrial structure. The first is the ratio of the main business income of the high-tech industry to the main business income of industrial enterprises above a designated size. The second is the ratio of the full-time equivalent of R&D personnel to the employees of industrial enterprises. The index of the advanced industrial structure is defined as the arithmetic average of these two ratios. The effects of TLT on economic growth may vary according to the different development intervals within an advanced industrial structure. To test this assumption, we set the advanced industrial structure as the threshold variable and analyze this threshold effect.

After threshold model testing, the estimation results in Table 9 indicate the existence of a single threshold effect in the advanced industrial structure. Therefore, using different threshold values associated with an advanced industrial structure, this paper establishes dummy variables and generates cross-multiplier terms with TLT variables. This enables an investigation of the nonlinear relationship between TLT and economic growth. The specific form of the threshold model is as follows:

$$
\begin{aligned}
ln(pgdp)_{it} = \ & \alpha_o + \alpha_1 ln(industry)_{it} + \alpha_2 ln(govern)_{it} + \alpha_3 ln(edu)_{it} + \alpha_5 ln(peo)_{it} + \alpha_6 ln(invest)_{it} + \alpha_7 ln(trade)_{it} + \\
& \alpha_8 ln(elder)_{it} + \alpha_9 ln(tfp)_{it} + \beta_1 ln(land)_{it} \cdot I(ad - ind - str_{it} \leq \gamma_1) + \beta_2 ln(land)_{it} \cdot I(ad - nd - str_{it} > \gamma_1) + v_{it}
\end{aligned}
\tag{11}
$$

The threshold effect test results in Table 9 show that the advanced industrial structure *ad-ind-str* is divided into two intervals: *ad-ind-str* $\leq$ 0.0880 and *ad-ind-str* > 0.0880. The impact of TLT on economic growth is then analyzed through the lens of the advanced industrial structure in different sections. Table 8 shows the regression results of these two variables in different threshold intervals.

The results in Table 10 indicate that TLT has positive impacts on economic growth in the different threshold intervals. This positive effect is gradually strengthened as the advanced level of the industrial structure increases. In the first threshold interval, the advanced industrial structure is at a relatively low level, characterized by large volumes of traditional industrial products and a high transportation volume. Both depend highly on transportation infrastructure. Therefore, TLT plays a vital role in promoting economic growth. In the second threshold interval, as the advanced industrial structure continues to develop, a new type of strategic force gradually emerges, represented by the high-technology industry. The high-end service sector, represented by 5G, artificial intelligence, and big data with 5G, powerfully combines with the high-end manufacturing industry, represented by high-speed rail technology. This innovates the traditional transportation system and leads to an advanced intelligent transportation system, enhancing efficiency and service level of transportation. As a result, the economy achieves effective growth.

## 5. Discussion

### 5.1. Spatial Spillover and Time-Lag Effects of TLT Jointly Promote Economic Growth

Previous studies have focused on the spatial and temporal patterns of transportation land use in China and its influencing factors but have not examined the coordination and adaptation of transportation land use and economic development level [50]. Using data about provincial transportation land use from 2007 to 2019, this study examined the spatial spillover and time-lag effects of TLT on economic growth and the mechanisms involved in the impacts.

The results show that economic growth is significantly affected by the spatial spillover and time-lag effects of TLT. From the perspective of spatial interaction, as the national economy rapidly develops, transportation land serves the local economy and has an increasingly close connection between regions. Production factors, including technology, talent, and resources, flow more smoothly through transportation, releasing the spatial spillover effect. From the perspective of changes over time, transportation land promotes economic growth both during and after construction. The needs of economic and social development can be better satisfied by the supply of transportation land. However, transportation land construction takes a long time to build, operate, and transfer. The construction process drives employment and enhances industrial interactions, increasing consumption and strengthening the transmission of the industrial chain. After the construction process, transportation accessibility improves, elevating the efficiency of urban economies.

### 5.2. Stage of Economic Development, Industrial Structure, and Urbanization Level Have Significant Threshold Effects on TLT to Promote Economic Growth

Previous studies found that the economic development stage, industrial structure, and urbanization level affect TLT [74]. However, few studies have investigated their threshold effects on the promotion of economic growth by TLT. This study found that transportation land has a nonlinear effect on economic growth, influenced by economic development

stage, industrial structure, urbanization level, and other factors. The stage of economic development has a significant heterogeneous effect on transportation land in promoting economic growth. In regions at the primary, middle, and late industrialized stages, the role of transportation land in promoting economic growth remains high. However, in regions at a primary development stage, its effect is stable at a low level. As a possible cause for this result, multistage regions are in a stepwise stage of industrial development and urgently need transportation land to meet productive economic activities.

Many developing countries are facing the risk of shortages in transportation land supply and the challenge of a lack of effective urban planning [75]. For example, in Pakistan, areas with less transportation infrastructure have higher household transportation expenditures [76]. In the leap from the second stage to the third stage, the primary development stage led to a qualitative breakthrough in the demand and supply of transportation land. Economic development demands have transformed from increased quantity to improved quality, changing the target supply and development mode of transportation land.

Turning to a developed country, the self-reinforcing model adopted by the United States over the past few decades has adapted to a growing dependence on automobiles, car-oriented planning and development, and segregated and sprawling land use. This trend has negatively impacted the economy, society, and environment [74], indicating that developed countries, or countries in the later stage of industrialization, should transform their original extensive mode of transportation land supply to be more focused, renewing the original land, comprehensively planning for different types of land use, encouraging balanced land use, and realizing leap-forward development.

Industrialization and urbanization are two important aspects of economic development and provide two major research perspectives to observe the mechanism by which TLT promotes economic growth. When the proportion of tertiary industry exceeds 44.16%, the effect of transportation land on economic growth significantly increases. This may be because producer services included in the tertiary industry serve the manufacturing sector, achieving positive interactions and mutual enhancements. Producer services are independently differentiated from the service sector within the manufacturing industry and include the transportation, post, and telecommunications industries. This industry depends highly on transportation land. When advanced industrial levels exceed 8.8%, high-tech industry plays a prominent role in promoting the economic growth of transportation land. As a potential reason for this observation, high-tech industry development empowers transportation infrastructure and promotes its intelligent development with advanced intelligent manufacturing technology. Within the context of productive service industry empowerment and the high-tech industry leading advanced industrial development, TLT multiplies their role in promoting economic growth. When the urbanization rate exceeds 53.5%, the effect of transportation land on economic growth is also significantly increased, possibly because the urbanization process includes land urbanization and population urbanization, which both need transportation infrastructure construction, supported by transportation land. Transportation land can shape the layout of urban spatial structure, and transportation infrastructure can meet the needs of population mobility and agglomeration. When the level of population urbanization is high, the large population, high agglomeration, and high intensity of economic activities maximize the efficiency of transportation land use, accelerating its role.

*5.3. Limitations and Future Directions*

This paper focuses on the temporal and spatial effects and the mechanism by which TLT impacts economic growth and analyzes the nonlinear effects of the economic development stage, industrial structure, urbanization level, and other factors on TLT in promoting economic growth. However, there are many types of transportation land, including land for railways, highways, airports, ports and wharfs, pipeline transportation, urban rail transit, urban roads, transportation stations, and other transportation facilities. We studied the impact of the overall transportation land area on economic growth but did neither examine

the role of different transportation land types on economic growth nor the overall planning of these land types. Transportation land is an essential part of urban land and exerts an important influence on the development of the urban form and land-use patterns. There is a lack of comparative studies on the negative and positive effects of TLT and of judgment studies on the excess and insufficiency of transportation land supply levels. Therefore, future research should explore the role of different transportation land types on economic growth, the overall planning of different transportation land types, and their impact on urban morphology and land-use patterns. Furthermore, the economic benefits of TLT compared with negative excess should be considered in the future.

## 6. Conclusions

This study investigates the influence of TLT on economic growth, given the significant importance of transportation land. By decomposing the effect of TLT into the employment driving effect, industrial interaction effect, and efficiency improvement effect, the study demonstrates that TLT can theoretically promote economic growth. Based on the empirical analysis of China's provincial panel data from 2007 to 2019, we applied a spatial Durbin model, time-lag model, and threshold regression model to test the multidimensional relationship between TLT and economic growth. The key results are as follows. (1) TLT is a necessary condition for economic growth. This requires the continuous expansion of transportation land. Constructing transportation land between adjacent areas has significant spatial spillover effects. (2) With time-lag effects, TLT can promote economic growth by increasing employment and industrial interactions and by improving economic operation efficiency. (3) Transportation land has a significant heterogeneous moderating effect with respect to economic growth, with key differences based on economic development stage, industrial structure, and urbanization rate.

This study has important implications for developing countries and areas. First, continuous attention has to be paid to the supply and construction of transportation land. However, levels should be determined based on the upgrade path associated with the industrial structure. Different strategies will have different impacts on the function of transportation land. Second, the boundary of urban development should be scientifically planned with significant attention to land-intensive development. Ensuring that these assets continue contributing to urban growth requires changing the mode of economic growth to focus on technological progress and efficiency improvements instead of merely expanding urban construction land. Due to the large differences in land resources and economic development between regions, the government should determine development goals and paths based on local conditions. This should encourage economic development and realize a virtuous cycle of intensive land resource use and sustained and stable economic growth.

**Author Contributions:** Conceptualization, M.Z. and J.L.; methodology, M.Z. and H.L.; software, M.Z. and Z.L.; validation, M.Z. and Z.L.; formal analysis, Y.Z.; investigation, M.Z.; resources, M.Z.; data curation, X.W.; writing—original draft preparation, M.Z.; writing—review and editing, J.L. and H.L.; visualization, H.L.; supervision, J.L. and H.L.; funding acquisition, M.Z. All authors have read and agreed to the published version of the manuscript.

**Funding:** This research was funded by the "Theoretical and economic research-oriented innovation team" of the youth innovation talent introduction and education plan of colleges and universities in Shandong Province; "Effect of high-speed rail network on urban population distribution pattern in Shandong province" (20DRKJ02); Study on population flow in Shandong province under the background of "the snatch war" (19CJJJ13); Taishan Scholars Program (tsqn201909135); "Employment polarization effect of low-carbon development constraint from the perspective of unequal opportunities" (ZR2020QG040); Research on innovation of watershed ecological compensation system in water pollution prevention and control in Shandong Province (19CDCJ08).

**Data Availability Statement:** The TLT area data were obtained from the China Land Market Network. The explained variables and other control variables were obtained from the National Bureau of Statistics and provincial statistical yearbooks. The data of the industrial structure were sourced

from the China Industrial Enterprises Statistical Yearbook, the China High-tech Industry Statistical Yearbook, and the China Science and Technology Statistical Yearbook.

**Acknowledgments:** The authors thank the "Theoretical Economics Research Innovation Team" of the Youth Innovation Talent Introduction and Education Plan of Colleges and Universities in Shandong Province for financial support.

**Conflicts of Interest:** We declare that there are no conflicts of interest.

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
