# Peer review of "The Mechanisms of the Transportation Land Transfer Impact on Economic Growth: Evidence from China"

_land, doi:10.3390/land11010030_

Round 1

Reviewer 1 Report

China has a specific land tenure and tenure system that is very different from most countries globally. For this reason, the author should first explain this specificity. The mechanism by which land transportation is transferred is not well explained and is still difficult for readers from outside China to understand. Are all the lands referred to as land owned by the central government? What is the previous status of land use and tenure?

To whom is the land transferred? To the local government, the transportation project owner, or the other parties?

There needs to be a further explanation about land users and landowners as written in lines 14 and 15 (Abstract).

Why are the above tenure factors and previous land use factors not included in the framework as Figure 1?

In many developing countries, the development of transportation, directly and indirectly, impact the conversion of prime agricultural land, the conversion of lands that provide environmental services, and the marginalization of local communities. This study does not mention the above matters, how far the economic benefits of TLT compared to other negative excesses should be considered in the discussion chapter, and the limitations of the research scope.

From an economic perspective, apart from economic growth, we also need to consider the extent to which economic growth is in line with or contradicts the need to grow an inclusive economy.

MAUP (modifiable area unit problem) issue.

From the perspective of spatial analysis, to what extent does the choice of unit of analysis have an effect? This study chose to use the provincial unit. Is there a possible difference in the results using the city/district level analysis unit?

Author Response

Dear Editor and Reviewers,

Thank you very much for giving us an opportunity to revise our manuscript. We sincerely appreciate the editor and reviewers very much for the constructive comments and suggestions on our manuscript (land-1495828).

We have studied reviewers’ comments carefully. According to the reviewers’ detailed comments, we have made a very careful revision on the original manuscript. In addition to the reviewer’s suggestions, we also re-checked the manuscript and improved the written expression. All revised portions are marked in the revised manuscript which we would like to submit for your kind consideration. The specific amendments, responses and explanations are summarized as the following. We look forward to your response. Thank you so much.

Keep safe and take care always.

Yours sincerely,

The authors

Reviewer 2 Report

This is an interesting paper that examines the transportation land and economic growth in 30 provinces in China. The authors generate results based on the spatial Durbin Model, the time lag model, the meditation effect model, and the threshold effect analysis. The suggestions I have are intended to improve the paper further:

  1. The research question can be better clarified and mentioned in the first paragraph. Though the empirical study focuses on the provincial level in China, the research question should be motivated by a more general background. Therefore, some clarification on the difference between China and other countries, provincial level and other levels would better help the readers to understand the significance of the research and establish the link of this study to the existing literature.

  1. In terms of the spatial model, there is a need to further explain why to select the spatial Durbin model. When there are both spatial lag term and spatial error term, further comparison is requested to select an adequate model. Please refer to Elhost (2014) and Hui and Liang (2016). Besides, as it uses a spatial model, there should be more interpretation of the statistical result of the indirect effect.

  1. As to the mediation effect model, the author should clarify the specific model. If it is a regression model, there will be endogeneity that need to be solved or stated in the research limitation. Otherwise, it should apply structural equation modeling, which is widely used to tackle the mediation effect problems.

  1. The entire manuscript needs a thorough editing: (a) Some paragraph is unnecessary long, such as the explanation of variables can be combined and presented in a table. (b) The emphasis should be given to the methodology and research design rather than the models, and better before the paragraph of variables, so that the readers can have a clear and general picture of what steps the authors have followed to address the problems. (c) Please confirm if the “transportation land transfer” is an adequate translation. Does it refer to the newly transferred transportation land per year?

Author Response

(The authors gave the same response as above.)

Reviewer 3 Report

This study analyzes the impact of transportation land transfer (TLT) on economic growth. The study seems to be meaningful because it attempts to analyze the impact of transportation policy on economic growth. However, the following issues make the study difficult to understand.

First of all, TLT is not clearly defined, and the analysis is conducted without sufficient explanation as to why TLT would have an impact on economic growth. This makes it difficult to understand whether the regression analysis is a result based on a causal relationship or merely an indication of a correlation.

For equation (1), there is no explanation of Xit, and it is unclear what it refers to. Also, lnpgdp is probably the logarithm of pgdp, in which case it should be written as ln(pgdp) or lnpgdp should be explained precisely. The same is true for many other variables.

The X in Transportation land transfer(X) in Figure 1 seems to have a different meaning from the X in the equations. Since "land is the index of the TLT area", I guess "Transportation land transfer=land", but the description should be unified to avoid confusion. Similarly, is Economic growth (Y) pgdp? In that case, the definition of pgdp should also be more precise in what it represents.

Are the numbers in parentheses shown in Table 1 t-values? Shouldn't non-significant values be interpreted as a lack of clarity in the effect? Also, the R2 of the model is very low. Shouldn't it be clearly stated that the explanatory power of the factors taken into account for economic growth is negligible?

Why is the R2 of models 9 and 10 in Table 2 so low, but the R2 of model 11 in Table 3 so high? An explanation is needed.

In the caption of Table 4, it is necessary to describe what the numbers indicate.

For these reasons, Major revision is considered necessary.

Author Response

(The authors gave the same response as above.)

Round 2

Reviewer 3 Report

Most of the questions raised by the reviewers have been answered by the authors' revision of the paper, and many of the questions have been resolved. However, some of the corrections are considered inadequate and minor revisions are deemed necessary.

The variable in Eqs. (1)-(8) is still lngdp, although it has been corrected to ln(gdp) in the table.

I still don't understand the meaning of using X and Y in Figure 1: Transportation Land Transfer (X) and Economic Growth (Y). The parameters in the figure correspond to the equations, but in equations (1)-(8), X is a control variable other than land, and Y is not used. It is unclear why it is necessary to rewrite them as X and Y. Shouldn't the notation correspond to the equation, e.g., land instead of X and pdgp instead of Y?

In the previous paper, I did not ask this question because I did not understand the meaning of Xit, but in equations (1)-(8), should the coefficient of Xit be 1? Looking at the table, it seems that the coefficient of Xit is also estimated. Also, Xit seems to represent a single control variable, but the table shows that there are multiple control variables. The equations need to be written accurately, such as in vector notation, so that it can be seen that multiple control variables are used.

The fourth term on the right-hand side of equation (1) is not clear about what it is summing over.

In line 313, it says "Xij is the set of control variables", but Xit and Xjt are used in the equation (1).

Author Response

(The authors gave the same response as above.)
